# Is the Relationship Between Cardiovascular Disease and Alzheimer’s Disease Genetic? A Scoping Review

**DOI:** 10.3390/genes15121509

**Published:** 2024-11-25

**Authors:** Anni Moore, Marylyn D. Ritchie

**Affiliations:** 1Department of Genetics, Perelman School of Medicine, University of Pennsylvania, Philadelphia, PA 19104, USA; moorea@pennmedicine.upenn.edu; 2Division of Informatics, Department of Biostatistics, Epidemiology & Informatics, Perelman School of Medicine, University of Pennsylvania, Philadelphia, PA 19104, USA; 3Penn Institute for Biomedical Informatics, Perelman School of Medicine, University of Pennsylvania, Philadelphia, PA 19104, USA

**Keywords:** cardiovascular disease, Alzheimer’s disease, dementia, genomic, gene

## Abstract

Background/Objectives: Cardiovascular disease (CVD) and Alzheimer’s disease (AD) are two diseases highly prevalent in the aging population and often co-occur. The exact relationship between the two diseases is uncertain, though epidemiological studies have demonstrated that CVDs appear to increase the risk of AD and vice versa. This scoping review aims to examine the current identified overlapping genetics between CVDs and AD at the individual gene level and at the shared pathway level. Methods: Following PRISMA-ScR guidelines for a scoping review, we searched the PubMed and Scopus databases from 1990 to October 2024 for articles that involved (1) CVDs, (2) AD, and (3) used statistical methods to parse genetic relationships. Results: Our search yielded 2918 articles, of which 274 articles passed screening and were organized into two main sections: (1) evidence of shared genetic risk; and (2) shared mechanisms. The genes *APOE*, *PSEN1*, and *PSEN2* reportedly have wide effects across the AD and CVD spectrum, affecting both cardiac and brain tissues. Mechanistically, changes in three main pathways (lipid metabolism, blood pressure regulation, and the breakdown of the blood–brain barrier (BBB)) contribute to subclinical and etiological changes that promote both AD and CVD progression. However, genetic studies continue to be limited by the availability of longitudinal data and lack of cohorts that are representative of diverse populations. Conclusions: Highly penetrant familial genes simultaneously increase the risk of CVDs and AD. However, in most cases, sets of dysregulated genes within larger-scale mechanisms, like changes in lipid metabolism, blood pressure regulation, and BBB breakdown, increase the risk of both AD and CVDs and contribute to disease progression.

## 1. Introduction

Cardiovascular diseases (CVDs) are the number-one cause of death worldwide [1]. CVDs cover a category of diseases affecting the heart and vascular system, which supply blood throughout the body, and most commonly include ischemic heart disease, stroke, and heart failure [2]. CVDs are often progressive and are highly prevalent among the aging population. The heritability (h^2^) of CVDs ranges from 30 to 50% [3] or up to 60% for coronary artery disease (CAD) [4], making genetic predisposition an important component of disease prediction.

Alzheimer’s disease (AD) is the most common form of dementia in the aging population, with as many as 11% of people over the age of 65 living with AD [5]. AD is characterized by the deposition of amyloid-β (Aβ) protein plaques around neurons and the appearance of tau tangles in the brain [6,7,8]. Biological changes that lead to AD have a substantial heritable component, where heritability is 58–79% [9]. Like that of CVD, AD progression begins without noticeable symptoms for 10–20 years prior to diagnosis. Some models of AD now predict that changes in the vasculature within the brain may occur before other more established biomarkers like the deposition of amyloid and tau [10], making the role of the vascular system in AD an important area of study.

The rise in epidemiological studies between CVDs and AD over the years has suggested that some relationship exists between these two complex diseases given the apparent influence of one over the other. As far back as 1997, Hoffman and colleges in the Rotterdam Study determined that patients with severe arteriosclerosis carried a two to three times higher risk of developing AD than those without arteriosclerosis [11]. Patients diagnosed with CVDs, like stroke [12] and atrial fibrillation, are more likely to later be diagnosed with dementia and AD [11,13,14,15,16,17,18,19,20,21,22] and have protein plaques in the brain [23]. An excess of cognitive impairment and eventual AD has also been demonstrated in patients with additional forms of CVDs, including heart failure (HF) [20,24,25]. Conversely, brain images from elderly individuals have been used to accurately predict heart disease risk [26]. In combination, cases with both CVD and AD experience worsened outcomes, experiencing more rapid cognitive decline than patients with either CVD or AD [27,28,29]. In recent years, cardiovascular health has been reported to be independently and linearly associated with cognitive function [20,30,31,32,33,34]. In other words, poorer cardiovascular health, through a buildup of cardiac dysfunctions, leads to neurodegeneration [35]. CVDs, such as coronary heart disease, heart failure, and atrial fibrillation, are now established risk factors of cognitive impairment and dementia [20].

With the rise in the availability of next-generation sequencing (NGS) and biobanks as public resources, teams have turned to genome-wide association studies (GWASs), genetic correlation, Mendelian randomization (MR), and genetically predicted biomarkers, amongst other statistical genetics methods, to provide clues towards the shared genetic risk and markers of overlapping mechanisms between diseases. Interest in the substantial overlap of CVD and AD diagnoses has led to an influx of studies looking for mechanistic overlap to better understand why these two prevalent and complex diseases often seem to co-occur. Given the high heritability of both AD and CVDs, along with their epidemiological co-occurrence, the hypothesis of shared genetic risk factors is probable.

Here, we conduct a scoping review to summarize and present a broad overview of the current knowledge of overlapping genetics and mechanisms between CVDs and AD. Specifically, the aim of the present study is to answer the following questions: (1) Are there specific genes that have widespread effects on the etiology of both CVDs and AD? (2) Are there mechanistic changes that occur in both CVDs and AD that dysregulate similar genes and pathways? To answer these questions, we aim to identify genes commonly associated with both diseases and characterize how these promote dysregulation in both brain and cardiac systems based on statistical genetics methods. We also identify mechanisms observed in both complex disease domains and how the involved genes fit into the pathways described.

## 2. Materials and Methods

To perform this review, we followed PRISMA-ScR guidelines for scoping reviews [36]. The focus of the current review was to identify key genetic concepts shared between CVDs and AD. It was specifically focused on articles that used statistical genetics methods to establish these overlaps.

As a preliminary search, papers were extracted from two databases, PubMed https://www.pubmed.ncbi.nlm.nih.gov/ (accessed on 24 October 2024) and Scopus https://www.scopus.com/ (accessed on 24 October 2024), matching the search results for (“Alzheimer’s cardiovascular genome”) OR (“Alzheimer’s heart cardiovascular genome OR genetic”). No limits were placed on the date or country of origin; therefore, papers published between 1990 and October 2024 were collected. This initial search identified a combined 2918 articles (Figure 1). A data extraction form in Microsoft Excel was used to collect information from each article, including the title, authorship, publication date, article type, journal, DOI, and PMID.

After pooling search results from all three keyword searches in PubMed and Scopus, duplicate articles were removed, leaving only one copy to be screened. Only articles that contained completed research analyses were included. All non-articles or preprints were excluded. This included reviews, book chapters, commentaries, conference papers, editorials, letters, notes, protocols, and short surveys. The title and abstract of each of the remaining 1058 articles were then screened by AM (Appendix A). Articles were included in initial abstract screening if they (1) described a performed study; (2) performed the study with human and/or murine samples; (3) focused on a CVD, AD, or both; (4) used statistical methods in their approach; and (5) were within the scope of the final topics covered. These five criteria were included as columns in the Excel sheet containing the information of all screened articles. A total of 216 articles passed the screening process and were included in the scoping review (Appendix A). Relevant articles from the reference lists of the included articles that also fitted our criteria were also considered and listed in the “Other Methods” section of the procedural flow chart. A total of 58 articles were included from this section (Appendix A). In total, 274 articles were included in the final scoping review (Figure 1).

In the following sections of this scoping review, we will explore the evidence for a shared genetic risk between CVD and AD. We will delve into the specific shared genes between the two disease domains; we will also describe their different proposed mechanisms and shared biology (namely lipid metabolism, changes in blood pressure, and blood–brain barrier impairment). Using a “gene-specific” lens followed by a “shared mechanism” lens allows us to explore the different approaches that researchers have used to address this question of an underlying relationship between AD and CVD.

## 3. Results

In total, we considered 274 papers published between 1990 and October 2024 (Figure 1). The papers were organized based on (1) evidence of an overall shared genetic risk and (2) shared mechanisms.

### 3.1. Evidence of Overall Shared Genetic Risk

Given the high heritability of both CVD and AD, it has been hypothesized that genetics play a role in bridging the relationship between the two diseases. Statistical genetics methods have identified sources of genetic variation linked individually to AD and CVDs [37,38,39,40,41,42,43,44,45,46,47,48,49,50,51,52,53,54,55,56,57], with some studies beginning to identify shared genetic changes observed in both diseases (Table 1).

Studies have found success using common genetic variants to find overlapping associations, with significant variants distributed throughout the genome. In a single GWAS, Broce et al. was able to identify 90 different genetic variants on 19 different chromosomes, jointly associated with increased AD and cardiovascular outcomes while excluding *APOE* [84]. In examining the relationship between CAD and AD neuritic plaques, Beeri et al. also concluded that there was a significant relationship between the two even when controlling for *APOE* genotype, suggesting that *APOE* may not be the only connecting factor [138]. By running a pathway analysis of an AD GWAS and brain expression results, Xiang et al. also noted that variants clustered into cardiovascular disease pathways including cardiomyopathies [66]. Several pleiotropic loci have been identified between AD and CAD [84,98,174]; for example, the genes *ARHGAP26* [84,99], *FMNL2* [175], *TMEM43* [91], *ABI3* [59], and *KIAA1462* [67,100,176,177] have all exhibited evidence of AD and CVD or vasculature involvement. Pleiotropy occurs when one variant or gene independently affects two different outcomes or diseases. Statistically significant results from AD GWASs [40,178] and transcriptomic studies have pointed towards genes involved in altered cardiovascular disease-related pathways [90,92]. Conversely, transcriptomics of cardiac tissues from patients with CVDs like dilated cardiomyopathy have shown enrichment for AD-related pathways [87].

Polygenic risk scores, which use genetic factors to predict disease outcomes, have been used to successfully predict both AD and CVD [34,41,78,86,128]. Additionally, using AD genetic risk to predict CVD outcomes and CVD genetic risk to predict AD has also proven successful, demonstrating the strong genetic underpinnings of the two diseases. For example, Zhang and colleagues found that genetic AD risk causally predicted an increased risk of angina pectoris [114]. Conversely, using CVD risk genetics to predict cognitive impairment has also been proven to be possible [104]. Specifically, atrial fibrillation genetic risk has been reported to be associated with dementia outcomes [109]. Cardioembolic stroke genetics successfully have impacted AD risk prediction [114]. Additionally, Kirby and colleagues found that of seven CAD traits tested (angina pectoris, cardiac dysrhythmias, coronary arteriosclerosis, ischemic heart disease, myocardial infarction, non-specific chest pain, and CAD) all were genetically correlated with AD [95].

However, not all studies have come to such supportive conclusions. For instance, when using a gene-based approach, Karlsson et al. determined that there was no genetic overlap between CAD and AD using summary statistics [102]. Using linkage disequilibrium (LD) score regression, Bulik-Sullivan also found no evidence of correlation [179]. The same conclusion was drawn using an MR approach when excluding *APOE* [96]. Zhong et al. likewise found no shared causality or genetic risk between AD and HF, ischemic heart disease, coronary heart disease (CHD), or atherosclerosis using MR analyses [113], though Chen et al. did find causality between cardiomyopathy and AD, as well as hypertension and AD, using MR [93]. Chi and colleagues also found significant causality between AD genetics and myocardial infarction [118]. MR mimics a randomized control trial, but instead of trial randomization, it uses the natural randomization seen in genetic variation in populations to prove the causality of effect alleles on an outcome. However, the genetic variation included is often limited to significantly associated disease variants determined from GWASs. This becomes a limitation if the source GWAS is underpowered or the input variants used in the MR analyses do not explain all variance in the diseases they represent; this may also limit the full interpretability of the conclusions. Given that pleiotropic variants have been found between CVD phenotypes and AD, which breaks a central assumption of MR, MR may not always be the best method to assess genetic connections. For more on MR, please review the articles from Emdin [180] or Sanderson [181].

Instead of inherent genetic overlap, it has also been hypothesized that CVD may act as an external stressor that worsens dementia outcomes. Patients with a higher genetic risk of CVD are also at a higher risk of dementia compared to patients with a lower CVD genetic risk [102]. When using genetic variants associated with brain cortical thickness, where a decreased thickness is associated with AD, these genetic variants differentially affect the measures of AD in the presence of modifiable cardiovascular risk factors [108]. This interaction between cortical thickness variants and cardiovascular conditions exerts additional risk beyond genes or environment alone [108]. In consideration of this argument that CVD somehow mediates or acts with AD pathology to make symptoms worse, Kobayashi et al. hypothesized that CVD may alter the methylation of genes which increases AD predisposition [69]. In their study, the authors noted that AD patients without CVD had higher levels of *COASY* methylation in blood than AD patients with CVD [60]. This hypothesis may also help explain the lack of positive associations seen in some analyses assessing direct genetic overlap. Genetic risk may predispose patients to AD or CVD, but the combination of genetic risk and environmental factors may also act in concert to propagate dysregulation.

As described above, there have been many studies looking at the shared genetics between AD and CVD, with some studies describing evidence of shared genetics while others disagree. Though many studies have looked for genome-wide associations, there is also a substantial literature exploring two specific genetic underpinnings of AD and CVD. In the next sections, we will go into more detail on those two specific gene sets: (1) *PSEN* genes and (2) *APOE*.

#### 3.1.1. Presenilin (*PSEN*) Genes

Mutations in presenilin (*PSEN*) 1 or 2 are some of the most penetrant genes of familial early-onset AD (<65 years) [182], with nearly 50% of early-onset AD cases having at least one *PSEN* mutation [183]. *PSEN* is expressed ubiquitously throughout tissues, including the brain and cardiac tissues, and believed to be involved in intercellular signaling and transport [183]. Regarding AD etiology, the exact mechanism of *PSEN* genes has still not been fully elucidated. However, it has been noted that *PSEN2* expression is upregulated in postmortem AD brain tissue [119] and leads to an increase in Aβ production [182]. This has been replicated in mouse models with mutant *PSEN1* or *PSEN2* alone, exhibiting increased Aβ levels and early vascular remodeling, potentially in response to Aβ deposition [120].

Within the vascular system, both genes are expressed in the heart and are critical to cardiac development and contraction [123,124,126,127]. In a clinical four-family study, one mutation in familial *PSEN1* resulted in the necessity of cardiac transplantation or death [127]. Moreover, mutations of *PSEN* genes were found in cases of idiopathic dilated cardiomyopathy and HF. The hearts of these patients contained amyloid protein aggregates similar to those found in AD [125,127,184]. In a separate study, glucose and oxygen deprivation were separately found to upregulate *PSEN2* expression up to 200% [125], both of which are mechanisms known to impact CVD and AD progression.

#### 3.1.2. *APOE*

*APOE* is a gene located on chromosome 19 and is expressed throughout the body. *APOE* is the most influential genetic risk factor for late-onset AD [61,130,138]. Three main isoforms, e2, e3, and e4, have been studied, each with differing effects on lipid and amyloid metabolism and inflammatory response [158,173,185]. Each known isoform has also been linked to varying impacts on the associated disease risk [149,186] and age of AD onset [41,135,137,170]. *APOE4* confers the greatest AD risk and is heavily associated with cognitive decline and dementia in genetic studies [44,111,128,129,135,187]. The cognition of *APOE4* carriers dramatically worsens in the presence of CVDs [149,151] and *APOE4* carriers demonstrate faster cognitive decline than other *APOE* variants [134]. The brains of human *APOE4* carriers and mouse models also have greater overall Aβ plaque loads and tau concentrations than other isoforms [161,164,188,189,190,191]. On the other hand, *APOE2* has been found to confer a protective effect on AD risk by delaying its onset [135,149,171]. *APOE2* carriers also have lower levels of Aβ deposition [161,192], though the *APOE2* isoform still increases the risk of CVD, especially in men [186].

*APOE* genotypes, especially *APOE4*, have similarly been associated with an increased risk of CVD phenotypes [158,193] including CAD, atherosclerosis, dilated cardiomyopathy [194], CHD [144], hypercholesterolemia [133,160], ischemic heart disease [133], coronary sclerosis [138,142], stroke [140], and cardiovascular mortality [136], though not universally [139,140,145,153,195]. Studies with *APOE2/3* isoforms have detected their protective effect for hypercholesterolemia, ischemic heart disease, and stroke, but *APOE2/2* revealed an increased risk of vascular disease, thromboembolism, and arterial aneurysm, indicating that genetic variations in *APOE* do play a role in CVD risk [133,140,159,196]. Grace et al. found that different genetic variants within the *APOE* locus accounted for its association with both CAD and late-onset AD when performing MR causal analysis [96]. AD association peaked at the *APOE4* locus (rs6857), along with plasma low-density lipoprotein (LDL) cholesterol and cortical amyloid load [96]. The variant rs4420638 in the LD region of *APOE2* was also significantly associated with LDL cholesterol but not with cortical amyloid [96], suggesting that *APOE4* may play a greater role in amyloid metabolism, while *APOE2* primarily affects lipid metabolism. In another study by Selvaraj et al., *APOE4* did not seem to be associated with cardiac structure, function, or biochemical markers of HF [143]. Versmissen and colleagues also concluded that LDL receptor function is essential for the detrimental effects of *APOE4* on CHD risk [145], further pointing to lipids as the main connector of *APOE4* to CVDs.

*PSEN* genes and *APOE* have become well-established risk genes for both AD and CVD. Mutations in either gene family effectively determines the timeline of AD onset and substantially increases the likelihood of adverse cardiac events. While much focus has been given to these two families of genes to better understand their impacts on the cardiovascular system and the brain, most studies investigating the commonalities between CVDs and AD have worked to identify the actual mechanisms that lead to the progression of both complex conditions. In the next section, we summarize the pathways believed to simultaneously contribute to the advancement of CVDs and AD, as well as the genes involved.

### 3.2. Shared Mechanisms of AD and CVD

In addition to co-occurrence, the appearance of CVDs seems to influence the frequency of AD and vice versa. This suggests that, etiologically, dysregulation within one comorbidity overlaps with and advances the other through shared mechanisms. Several underlying mechanisms have been proposed between CVDs and AD using shared genetic association and transcriptomic studies in vascular tissue, as summarized in Figure 2. For example, despite Aβ plaques and tau tangles as the signature hallmarks of AD pathology, these protein aggregates also appear in the vasculature and cardiac tissue of those with AD, along with cognitively healthy individuals with CVDs [13,184,194,197,198,199,200,201]. Like in the brains of AD patients, these deposits have consequential effects, impairing tissue function over time [13,117,194]. This initial deposition is in part mediated by genetic changes in blood pressure and cardiac function which impact Aβ clearance [107,116,202]. Genetic association studies of structural and functional cardiac measures have found significant association of genes, including *TOMM40* and *BIN1*, which similarly associate with AD risk [82,203,204,205,206].

Prolonged changes in blood pressure regulation impacts vascular integrity, initiates a widespread inflammatory response, and precipitates the breakdown of the blood–brain barrier (BBB), which is essential for managing safe molecular exchange between brain parenchyma and circulating blood proteins. The investigation of BBB breakdown using the epithelial cell transcript measures of the vascular tissue of the AD brain compared to vessel damage from CVD patients with aortic stiffness or atherosclerosis, demonstrated an overlapping downregulation of angiogenic genes, like *VEGFA* and *IGF1* [83,207,208,209,210,211], and upregulation of inflammatory markers such as *MMP9* [212]. These processes can be further exacerbated by genetic and environmental changes in lipid metabolism, whereby increasing levels of lipid accumulate in vessels along with proteins. The isoforms of *APOE* can partially account for these effects. In the following sections, we discuss how genetics and gene expression changes advance disease progression through the shared mechanisms mentioned here. We describe three main pathways observed in both AD and CVDs: (1) altered lipid metabolism (Table 2), (2) blood pressure regulation (Table 3), and (3) BBB impairment (Table 4).

#### 3.2.1. Altered Lipid Levels and Metabolism

Cholesterol is one of the most established risk factors for CVD [213] and is often measured in its two forms: low-density lipoprotein (LDL) and high-density lipoprotein (HDL). Increased levels of cholesterol have also been reported to increase the risk of atherosclerosis [214], myocardial infarction, CHD [215], and stroke [216,217,218], as excess cholesterol particles inevitably build up in vessels and arteries and block blood flow. Genetic studies have linked measurable and genetically predicted lipid levels with CAD [219,220]. Specific genes with impacts on cholesterol levels have also been detected. For example, the differential methylation of *ABCG1* has been associated with HDL and triglyceride (TG) levels in coronary heart disease [221]. TGs are another form of lipids measured in blood. Statins have been proven to effectively lower LDL cholesterol and reduce adverse cardiac events [222,223].

However, cholesterol has impacts beyond the cardiovascular system. The brain is the most cholesterol-rich organ of the body, with 50% of the brain’s dry weight attributed to lipids [224], and high cholesterol has also been implicated in AD [209,225], including through genetic overlap [84,102,209,226,227,228]. Causality and correlation analyses have further linked LDL [59,229,230] and HDL [59,231] to AD. *APOE*, along with *APOJ* or *CLU*, *ABCG1*, *APOA4*, and *ABCA7*, which are known to be involved in extracellular and intracellular cholesterol transport, are significantly associated with AD [57,64,84,132,232,233]. *APOJ* and *APOE* also appear to be upregulated in accordance with Aβ accumulation in the AD brain [234]. Other genes, including *BIN1*, *SORL1*, *PICALM*, which meditate cholesterol intake within cells, have also been described in AD [47,48,132]. Variants in the genes *HS3ST1* and *ECHD3* were explicitly found to be pleiotropic between AD and TG [228]. Statins, previously mentioned for their use in lowering cholesterol in CVDs, have also been used in clinical trials within AD patients with mixed effects [235].

Thus, it has been hypothesized that it is through this altered lipid profile that CVD and AD are connected (Table 2). Karlsson and colleagues demonstrated that patients with higher genetic risk scores (GRS) for CAD were also more at risk of developing dementia than patients with low CAD GRS [102]. However, they also noted that the most significant variants for CAD included in their GRS were variants within important genes for lipid levels. They also reported that significant lipid genes clustered differently for CVD and AD, suggesting that this risk stems from independent susceptibility to lipid dysregulation rather than overlapping genetics [102]. Lipoprotein(a) (Lp(a)) is a type of fat in the body that can increase the risk of cardiovascular disease and stroke [236]. Causal genetic risk has also been observed for Lp(a) and CVDs (peripheral artery disease, aortic aneurysm, ischemic stroke, aortic stenosis [237], and large artery stroke [238]). Lp(a) levels have had mixed associations with AD [58,172,238,239,240].

**Table 2 genes-15-01509-t002:** Summary of literature regarding changes in lipid levels and metabolism observed in CVD and AD. (MR = Mendelian randomization; PRS = polygenic risk score; GWAS = genome-wide association study).

Topic	Area of Focus	Articles
Lipid genetics	AD	correlation [95], association [57,84,226,227,241], MR [229,230,231,237,238,242,243], PRS [244], pleiotropy [228,245], gene [232], systems [246]
	CVD	correlation [95], GWAS [216,219,220], gene [215,247], MR [237,238], gene [248], association [217]
Plasma lipid levels	AD	[209,225,244,249]
	CVD	[56,218,221,236]

*APOE*, as previously discussed, seems to account for most of the variation studied in altered lipid profiles within these two complex diseases. Wu and colleagues found *APOE4* was associated specifically with “bad” cholesterol, including LDL, TG, and total cholesterol, and lower “good” cholesterol like HDL [58]. The same was observed in the case of the plasma measures of LDL, TG, and total lipids in a study by Karjalainen et al. [159]. In both studies, *APOE2* had a protective effect on LDL and HDL and total lipids [58,159]. Genetic enrichment of AD variants has been seen for LDL, HDL, TG, and total cholesterol even when excluding *APOE*, *HLA*, and *MAPT* LD regions [84]. However, Tan et al. concluded that plasma total cholesterol levels were not associated with AD incidence over the course of an 18-year follow up study [249]. Meanwhile, increased HDL levels in midlife have been associated with lower AD risk [245]. This may suggest that specific cholesterol types rather than overall cholesterol levels may play an important role.

*APOE* is also explicitly associated with increased risk of both AD and multiple CVDs. Multiple studies have implicated variants within *APOE* in the genetic association of all three: lipids, AD, and CVDs [62,95,157,250]. Patients with an *APOE4* variant have a higher risk of developing AD, CVD, and higher cholesterol levels compared to those without [141,166,251,252]. The impact of changes to the methylation of the *APOE* in relation to lipid and CVD risk has also been investigated [147,157]. Ji et al. reported hypermethylation of the *APOE* gene in coronary heart disease cases compared to controls, though both Karlsson et al. and, more recently, Mur et al. disagreed [131,146,157]. Through genetic correlation and gene-based testing, Kirby and colleagues determined that significant association of genes within the *APOE* region including *APOE* and *TOMM40* were shared between CAD traits, AD, and lipids, though causality analysis could not significantly connect AD with CAD traits or AD with lipids in any direction [95]. Specifically LDL, TG, and total cholesterol showed significant genetic correlation for AD and CAD separately but ultimately were not causal, indicating that another mechanism may exist connecting lipids to AD through CAD traits [95]. It has also been argued that the LDL receptor is more important to CHD risk than *APOE4.* As observed by Versmissen and colleagues, despite also having two copies of *APOE4*, patients with loss-of-function mutations in the LDL receptor locus, actually had a lower risk of CHD [145]. The same was true even in patients with a family history of hypercholesterolemia, indicating that LDL receptor function has a more direct effect on CHD risk than *APOE*.

Other genes are also believed to impact the lipid metabolism pathway in both AD and CVD etiology beyond *APOE*. The gene *FABP2*, implicated in TG synthesis , is associated with AD and cerebrovascular disease. *FABP2* was downregulated in the plasma of AD cases [209], and certain polymorphisms of the gene were more frequently associated with transient ischemic attacks and non-cardioembolic infarction [79]. Using bioinformatics methods, *GPBP1* was identified as overlapping between the two diseases [89]. *GPBP1* is believed to be involved in cholesterol metabolism and was observed to be downregulated in the vascular tissue of animal models exposed to hypercholesterolemia [253]. It has also been observed to be dysregulated in the cortex tissue of AD brains, with the authors noting differences between male and female samples [254]. The same study by Lee et al. also discussed *SETDB2*, which has links to lipid metabolism through glucocorticoids [255]. *SETDB2* is upregulated in atherosclerotic lesions [256] and has been associated with neuroinflammation in another study [257].

Another major proposed genetic risk factor overlapping between AD and CVDs is *PCSK9*. *PCSK9* plays a significant role in LDL cholesterol metabolism, breaking down LDL receptors before they can import cholesterol into cells. *PCSK9* mutations increase serum levels of LDL cholesterol [258]. In autopsy-confirmed late-onset AD cases, Picard et al. reported gene expression increases of *PCSK9* in the cortical brain and increased protein expression compared to age-matched controls [232]. This trend has also been seen in blood plasma [209] and cerebrospinal fluid (CSF) of AD cases [259], where CSF *PSCK9* protein levels were positively correlated with AD biomarkers including Aβ, pTau, and total tau, most notably in women [232,259]. *PCSK9* inhibitors, such as evolocumab, have been reported to successfully lower cholesterol levels in both diseases [242,248,260]. Williams et al. investigated the use of *PCSK9* inhibitors in AD using genetics, using variants within the *PCSK9* gene region to predict AD risk using MR [242]. In their analysis, *PCSK9* inhibitor genetic targets causally increased the risk of AD, though the authors admit that the inhibitors tested (evolocumab and alirocumab) are not able to cross the BBB and access the brain, so genetics may not be the most informative method to answer their question [242]. *PCSK9* inhibitors have however been observed to decrease the rates of cardiovascular events in genetic CVD models and in patients with known cardiovascular disease [248,260].

Furthermore, *PCSK9* stimulates the degradation of *BACE1* [261], which is a rate-limiting enzyme in the production of Aβ peptides via the cleavage of amyloid precursor protein (APP) [262]. Because of the neurotoxicity of Aβ in AD, *BACE1* has been a target of clinical trials. However, due to the enzyme’s low specificity for APP, these have remained unsuccessful for fears of off-target effects [263]. Additionally, netrin protein receptor DCC, which has also been identified as a *BACE1* target, plays a role in axon guidance and angiogenesis, impacting both neuronal and vascular pathways in AD brains [263]. The increased activity of *BACE1* in AD cases has also been related to the deposition and accumulation of Aβ in cerebral blood vessels [264,265]. Staining for Aβ peptide in vascular and cardiac tissue, Greco et al. linked the dysregulation of *BACE1* with the accumulation of Aβ in the heart [262]. The enzyme was specifically upregulated in the left ventricle (LV) of HF cases [262,266,267,268]. *BACE1* has also been discussed for its additional independent role in cardiomyocytes, where its interaction with KCNQ1 is necessary for cardiac muscle’s return to a resting state after contraction [269].

Many genes, including *APOE*, *PCSK9*, and *BACE1*, as discussed in this section have been identified for their involvement in lipid synthesis and transport in relation to AD and CVD progression. However, beyond their immediate effects, abnormal lipid metabolism has a greater effect downstream, and has been implicated in BBB integrity, APP processing, inflammation, and oxidative stress [243,244,270,271], which are all additional contributing factors to AD and CVDs [272].

#### 3.2.2. Blood Pressure Regulation

Blood pressure is a complex trait, though 30 to 60% of variance can be explained through genetics [76,273]. GWASs have successfully identified genes associated with both diastolic (DBP) and systolic blood pressure (SBP) [68,69,70,71,72,73,74,75,77,273,274], though the interaction of genes with the environment also accounts for a part of the variance explained [68,70]. Changes in blood pressure regulation can have a direct impact on CVD risk (Table 3). High blood pressure, or hypertension, can lead to a plethora of cardiovascular conditions, as the heart and vascular system wears down from constant pressure. Low pressure, or vessel blocks, can also have substantial negative consequences, preventing essential flow to vital organs. Greater focus has been given to the idea that subclinical changes in blood pressure [229] inmidlife [106,275,276,277,278] have impacts on the early stages of neurodegeneration and dementia. The regulation of blood pressure to the brain is extremely important for maintaining homeostasis, as high blood pressure can lead to cerebral tissue compression [279]. In the heart, insufficient pressure can lead to ischemia and infarction [279]. Both scenarios have been implicated in the etiology of AD progression.

**Table 3 genes-15-01509-t003:** Summary of literature describing changes in blood pressure and cardiac function and resulting impacts on AD and CVD progression. (GWAS = genome-wide association study; MR = Mendelian randomization).

Topic	Area of Focus	Articles
Blood pressure	AD	GWAS [68,69,71,72,73,74,75,76,273], methylation [70], MR [77,165,229,231,274], association [275,280]
	Hypertension (ACE)	CVD [281,282], AD [85,241,283,284,285,286,287,288,289,290]
	Lower cerebral blood flow	[82,291,292,293,294,295]
Impaired cardiac function	AD	[194,269,296,297,298,299,300]

##### Hypertension

Hypertension is both an established contributor to cardiovascular disease and an independent risk factor for cognitive decline and dementia [280,301]. Chronic high blood pressure on vessel and artery walls causes damage over time that predisposes patients to heart disease, stroke, and myocardial infarction. Both high DBP [13,117,194,197] and high SBP [111,231] have also been associated with cognitive impairment and AD risk through clinical and MR genetic analyses [111,112,231] (Table 3). Variants within the genes *APP* [101,296], *ACE* [281,283,284,288], and *APOE* [137,150,157,165,166,168] have all been implicated in both AD and CVDs. For example, *APP*, as previously discussed for its relationship with *BACE1,* is another highly penetrant, familial AD-causing gene [302]. In addition to its prevalent association with AD, different genotypes in variants within *APP* have also been observed to associate with differential hypertension risk. In a study focusing on the genotypic distributions of these APP variants in the Chinese population, patients with the CC genotype of rs2211772, had a decreased risk of hypertension compared to those with TT or TC at the same locus [101]. The same study also considered the effect of the methylation of the *APP* gene and noted that methylation levels at different CpG sites also correlated with differential hypertension risk [101].

The angiotensin-converting enzyme (*ACE*) gene has also been associated both with increased DBP and SBP through clinical studies and GWAS methods [85,241,281,285] (Table 3). *ACE* acts by initiating the renin–angiotensin system by converting angiotensin I to angiotensin II. Angiotensin II (ANGII) works downstream to induce the vasoconstriction of blood vessels and increase blood pressure through the retention of bodily water and sodium. A chronic increase in ANGII induces oxidative stress-response genes, resulting in damage to vulnerable organs including cardiovascular tissues, even when blood pressure tends to return to normal [281,303]. It causes a decrease in energy metabolism, as seen in damaged heart tissues and HF [304,305]. It also increases cardiac tissue thickening and stiffening, suggestive of cardiac hypertrophy [305]. A pathway analysis of differentially expressed genes in the hearts of mice injected with chronic ANGII showed evidence of significant associations with numerous AD and neurodegenerative disorder pathways [281]. The same patterns were found in the transcriptome profiling of hearts with end-stage dilated cardiomyopathy [88].

The use of *ACE* inhibitors has been a popular way ameliorate to blood pressure increases to treat hypertension and therefore to lower cardiovascular risk [285,286]. Scientists have also noticed additional beneficial effects of *ACE* inhibitors on dementia risk, and these inhibitors have since been tested in AD cohorts with mixed results [287,289,306]. In mice with AD, the centrally active *ACE* inhibitor, captopril, decelerated the accumulation of Aβ plaques and lowered *ACE* expression in the brain in a positive manner [288]. The same conclusion was also replicated in humans where *ACE* inhibitors delayed the onset of cognitive decline [290]. Further studies have suggested a role of *ACE* in Aβ levels and AD risk independently of *APOE* [283].

There are two main hypotheses as to how *ACE* affects both AD risk in the brain and blood pressure beyond the brain. The first involves the variants rs1800764 [282] and rs4291 [307], which are thought to increase *ACE* expression in serum, thereby increasing arterial hypertension. Simultaneous increased cerebral *ACE* has been detected in patients diagnosed with AD and further promotes neuroinflammation and attenuates cerebral blood flow, which may contribute to poor cerebral clearance [308]. In this hypothesis, the presence of Aβ plaques in the AD brain further up-regulates *ACE* expression and additionally triggers further angiotensin II-mediated Aβ generation [288]. A second hypothesis involves the *ACE* variant rs4308 [85], which has been explicitly found to be pleiotropic with an increased risk of AD and decreased DBP and SBP measures. The increased risk of AD was hypothesized to be mediated by decreased *ACE* expression in the brain, while increased *ACE* expression in the transverse colon and kidney was believed to explain the the effect on BP [85]. Bone et al. hypothesized that this increased expression of *ACE* in the kidney, a key mediating organ of blood pressure, slows the entire renin–angiotensin system, decreasing blood pressure [85]. Given *ACE*’s ability to cleave Aβ42 protein to Aβ40, a less pathogenic form, its downregulation in the brain results in an increase in the aggregating Aβ42 protein form and an increase in AD risk [309,310]. Thus, in this hypothesis, cerebral *ACE* expression is actually beneficial [286].

*APOE* and *APOE4* carrier status also plays a role in blood pressure [103,165,166]. AD patients with an *APOE4* allele are at a higher risk of structural cardiac changes such as LV disease in the early stages of AD progression [148] and display a greater maximum LV wall thickness [194], which may be a marker of chronic high blood pressure. These cardiac changes may help further explain the association of blood pressure and AD, though this association may also be in part due to *APOE*’s role in cholesterol metabolism whereby the buildup of lipid and protein plaques throughout the vasculature restricts blood flow, increasing blood pressure. Chronic high blood pressure also results in the increased circulation of toxic proteins [311,312,313] and harmful pressure being pushed into the cerebrovasculature of the brain [314], further exacerbating the inflammatory response, oxidative stress pathways, and overall damage as demonstrated by changes regional brain volumes [103]. Comparing cohorts of cognitively impaired and healthy individuals, Ngwa and colleges found that higher SBP was associated with decreased hippocampal volume [103], a brain region important to AD.

##### Reduced Cerebral Blood Flow

Infarctions, including heart attack, where blood is impaired from the heart, or stroke, where blood is blocked from the brain, are major cardiac events with life-threatening consequences. The blockage or lessening of blood flow to any organ, but especially the brain, can cause severe damage [315]. Both aortic stiffening and cerebral microinfarctions are common in aging brains and reduce blood flow to and around the brain [294]. Cardiac dysfunction and decreased cardiac output, which also limits the blood that reaches the brain, is directly associated with cognitive decline and AD models [168,292,296]. Increased diastolic blood pressure has also been linked to lower cerebral perfusion [293].

One potential contributing factor towards the lack of cardiac output or lack of blood to the brain is the impairment of cardiac structure and function [291]. LV structural traits like wall thickness and left atrial systolic dimension have substantial genetic components (h^2^ = 31%) [80,81] and have been studied in the context of CVDs like HF. At the cellular level, *TOMM40* encodes a subunit of a complex on the outer mitochondrial membrane essential for importing protein precursors and therefore vital for cellular energy production [203]. In cardiac muscle, impaired energy production affects overall cardiac impulse and contraction needed to properly pump blood throughout the body. Mutations in *TOMM40* are associated with a higher incidence of left bundle-branch block and indicative of defective electrical impulses [82]. This has been noticed in patients with atherosclerosis and results in a higher incidence of CVD and CVD related deaths [82]. Levels of plasma LDL and TG have also directly been associated with *TOMM40* [247,316], suggesting that poor heart function along with the buildup of lipid plaques in vessels may work synergistically towards poorer outcomes. Other cardiovascular risk factors like hypertension, diabetes, or unhealthy lifestyle habits like drinking, smoking, and inactivity may also work with downstream effect of *TOMM40* variants to exacerbate risk [317]. In addition to *TOMM40*, another gene has been hypothesized to impact heart function. Genetic instruments in AD were used to successfully predict unstable angina, when blood flow is poor through the heart [97]. A variant near *BIN1* was believed to link the two diseases within the analysis. In plasma, *BIN1* levels may also be used to indicate microvascular dysfunction in cardiomyocytes [206]. *BIN1* is necessary for calcium signaling and contraction in cardiomyocytes [205] and, likewise, for calcium signaling of neurons in the brain [318].

Poor cardiac function includes weaker pumping ability, meaning less blood is pumped out toward essential organs like the brain. Impaired cardiac contraction and a reduced left ventricular ejection fraction (LVEF) results in lower cardiac output, all of which have been associated with AD and dementia [194,299,300,319]. Lower cardiac output has negative effects on multiple necessary processes for maintaining brain homeostasis, as reliable blood flow and arterial pulses are required for glymphatic clearance in brain [202]. Decreased blood flow to the brain results in buildup of waste and miscellaneous proteins including Aβ and hyperphosphorylated tau (hTau) [107,116,202]. Decreased blood flow may also contribute to the initial stages of aggregating tau phosphorylation by destabilizing unphosphorylated tau [320] and inhibiting dephosphorylation [297,321]. This leads to an increase in the production to hTau, in addition to impaired clearance. Even in participants with healthy cognition, a lower LVEF was associated with the greater circulation of tau (t-tau and p-tau) [297], reinforcing the understanding that Aβ and tau aggregation and circulation begin decades before AD or CVD onset. Compromised blood flow to the brain additionally contributes to oxidative stress and inflammation, which is a large part of AD and CVD etiology [322,323].

The buildup of toxic proteins also contributes to the dysregulation of cerebral vasculature. In healthy brains, vessels constrict in instances of high pressure to mediate blood flow which initiates a negative feedback loop to finetune and re-dilate vessels appropriately. However, the cerebral arteries of APP mouse models (with overexpression of Aβ) have impaired vasodilation in response to increased blood pressure [324]. In AD and dementia models, these vessels do not re-dilate, and instead are stuck in a hyper-constricted state [169,324], reducing the overall blood flow allowed into the brain. Even in cognitively healthy individuals, higher blood pressure is associated with lower cerebral perfusion [293]. Similar conditions are also observed in cases of stroke, where vessel constriction decreases cerebral blood flow after an event [295]. Thus, hypertension may also conversely contribute to the obstruction of blood flow as symptoms worsen over time. The inhibition of *TMEM16A* expression within pericytes that make up vessel walls and are part of the BBB has been proposed as a method to reverse this constriction [295].

Many of the mechanisms described here appear to be disproportionately prevalent in *APOE4* carriers specifically. *APOE4* carriers have decreased cerebral blood flow velocity and demonstrate faster cerebral blood flow decline compared to non-*APOE4* carriers regardless of AD status [148,150,167]. *APOE4* carriers with increased vessel stiffness had a 10-fold increase in the deposition of cerebral Aβ compared to non-carriers with increased vessel stiffness [202]. They also have worse cognitive function when experiencing a reduction in cardiac output [152,168,292], though conclusions are more mixed [168]. Similarly, negative associations between heart size measures and AD protein aggregate markers in the brain detected by Beeri et al. suggest that more AD pathology is present when the heart is of a smaller size [138].

As it relates to dementia, *TOMM40* is a well-established risk gene for late-onset AD and resides near *APOE* [49,204,325,326], and differential *TOMM40* expression has been demonstrated in the brain [203]. In a clinical study examining the effects of polymorphism length at rs10524523, there was a dose-dependent increase associated with decreasing gray matter volume in the regions of the brain affected by late-onset AD [204]. The polymorphism length of the same *TOMM40* variant has been reported to impact the age of AD onset [326], though the proximity of *APOE* has resulted in some speculation of the association of *TOMM40* with AD [327]. Altered mitochondrial function and decreased oxidative function is similarly a hypothesized factor of AD etiology, and the altered expression of mitochondrial genes has been noted in frontal cortex tissue samples from AD patients [63,121].

Overall, the relationship between cardiac output and future dementias is difficult to study without considering the full timeline, including the many years before cardiac or brain symptoms of disease appear. Some of the aforementioned genetic studies may be limited as they are often not longitudinal studies or do not consider stratifying by age groups. Given that AD presents later in life, many AD cohorts often only include elderly participants. However, as it relates to this section, these cohorts would be missing essential midlife progressive changes to cardiac function. These cohorts would not reveal these midlife correlations, especially given that late-life vascular disorders do not seem to be correlated with dementia [105]. Future studies would benefit from multi-decade longitudinal observation to more deeply explore some of these proposed mechanisms.

#### 3.2.3. Blood–Brain Barrier Impairment

The BBB is essential for maintaining brain metabolism by regulating blood flow and the uptake of essential nutrients [279]. It exists as part of vessel walls and is mainly made up of epithelial cells that line inside the vasculature, along with astrocytes and pericytes. Within endothelial cells, the breakdown of the BBB is propagated by hypercholesterolemia [328], high pulse pressure [329], Aβ deposition [234,330,331], inflammation [332], and the aging microenvironment [333] (Table 4). These factors also propagate oxidative stress [334,335], worsening overall conditions. Eventually, a full barrier breakdown is marked by the loss of blood vessel density, the dysregulation of angiogenesis, and damage to the extracellular matrix, all of which contribute to and are exacerbated by Aβ levels [83,94,246,333]. This breakdown then allows the spread of built-up neurotoxic proteins throughout the circulatory system and the diffusion of toxic plasma proteins into the brain.

**Table 4 genes-15-01509-t004:** Summary of literature outlining nature of blood–brain barrier impairment in AD and CVDs. (Aβ = amyloid-β; CSF = cerebrospinal fluid).

Topic	Area of Focus	Articles
Blood–brain barrier function	General	systems [83,212,332], association [336], serum [337], RNAseq [333,338], CSF [339,340,341], microarray [342]
	Endothelial and pericyte cell dysregulation	RNAseq [207,210,333,343,344], systems [271,324,345,346]
	Dysregulated angiogenesis	[208,246,344,346,347]
	Aβ buildup and clearance	heart [13,18,200,262,348], brain [23,28,29,163,234,246,261,312,324,347,349,350,351,352]
	Tau buildup and clearance	heart [35], brain [212,341,347,351]
	Circulation of Aβ	[15,319,353,354]
	Circulation of tau	[197,201,314,353,355,356]
	Circulation of metabolites and proteins in AD	plasma [198,272,357,358,359,360,361,362], CSF [265], BDNF [363], homocysteine [364,365], uric acid [366]
	Circulation of metabolites and proteins in CVDs	BDNF [363], homocysteine [364,365], uric acid [366], plasma [272,360,366]

AD is characterized by increased BBB permeability, even in the subclinical stages of disease [339,340,342,367], resulting in oxidative stress and neuroinflammation [343]. The carriers of known AD risk marker, *APOE4*, have increased permeability compared to other isoform carriers [155] independent of Aβ and tau pathology [154]. *APOE4* also contributes to BBB damage in a dose-dependent manner [156,368] even in cognitively healthy individuals. Genetic variants associated with BBB integrity were even shown to be associated with changes in regional brain volumes [336].

##### Endothelial and Pericyte Cell Dysregulation

AD patients appear to have decreased endothelial and pericyte cell counts essential for maintaining the BBB, as well as altered gene expression in these cells types [338]. Pericytes play a role in regulating blood flow; thus, a decrease in pericyte cell counts and the dysregulation of remaining cells may affect cerebral blood flow, as described in the previous section [338,369].

The transcriptional profiling of endothelial tissues from the brains of AD patients has exhibited significant changes in the expression of genes and proteins necessary for maintaining cellular integrity and homeostasis, including AD gene markers like *APOE* [207] (Table 4). *APOE4* carriers specifically have had impaired signaling in pathways necessary for the regulation of vascular integrity [207]. There was also an increase in cellular senescence and apoptosis, alongside dysregulated pathways necessary for cerebral perfusion. Some of these changes can at least partially be attributed to Aβ. Intracellular Aβ, common to AD, has been shown to be cytotoxic to endothelial cells, disrupting essential endothelial cell survival pathways [330]. The clearance of Aβ from brain space into the vascular system via the BBB is impaired in AD [163,350], demonstrated by the downregulation of Aβ clearance pathway genes *PICALM*, *BIN1*, *CD2AP*, and *RIN3* in the vasculature of AD patients [207].

Additionally, expression of angiogenic regulators is necessary for the maintenance of blood vessels walls where the BBB is located. The altered expression of regulator genes is indicative of dysfunctional angiogenesis. Several effectors of angiogenesis, *VEGFA-VEGFR2*, *IGF1* [207,208,209,210,211,338], and the Ras signaling pathway [345,346,370,371], are all downregulated in cerebral endothelial cells of AD patients. The *VEGFA-VEGFR2* signaling pathway is also downregulated in pericytes [338] and *VEGFA* is decreased in plasma samples of patients with preclinical AD [347]. Other regulators, like *ANGPT2*, *FGF2*, *HIF1A*, and mTOR are all upregulated in AD [207,246,372,373]. The use of rapamycin to inhibit mTOR improved cerebrovascular density and cerebral blood flow in a mouse model of AD [372,373], indicating that this upregulation negatively contributes to conditions. *VEGFR1* is a substrate of *BACE1*, which was mentioned earlier for its role in the production of Aβ peptides [262,374]. AD cases with BBB dysfunction have low *BACE1* levels, though the proposed cause for this is not entirely clear [265]. Increased levels of angiostatin, a protein that prevents the growth of new blood vessels, was also found to be casual for increasing AD risk in an MR analysis of plasma proteins in dementia [357]. Altogether, these changes seem to reduce vascular density in the cardiovascular system of AD states [207,375].

Beyond impaired angiogenesis, further instability in vessel walls is caused by increasing tau accumulation and inflammation, which eventually leads to lower contractile function of vascular muscle. Low plasma *VEGFA* is associated with accelerated tau accumulation in the neocortex of individuals with elevated Aβ [347]. *ANGPT2*, when expressed too much, can promote vascular leak and instability [341,344,376,377] and is correlated with CSF tau levels and markers of vascular wall damage in early AD [341]. Similarly, *PDGF* is increased in vascular tissue in AD, specifically in pericytes, inhibiting vascular contractile genes and promoting the expression of neuroinflammatory markers like *MMP9* [212,338]. These changes are more extreme in the presence of tau and may contribute to a positive feedback loop of further tau phosphorylation [212]. *PDGF* is also released by pericyte cells as a result of stress and cell death, and an increased level of *PDGF* has been measured in patients with CVDs [378,379]. For example, in a study comparing *PDGF* receptor β (PDGFRβ) serum levels in patient groups with a history of stroke or ischemic attack versus without, patients with CVD events had significantly increased levels of PDGFRβ, indicative of BBB injury [337].

Many of the markers of endothelial dysfunction in AD such as *LOX1* have also been associated with CVDs like atherosclerosis [209,380]. High arterial stiffness is indicative of an increased risk of cardiovascular events and has a sizeable heritable component of 10–24% [83]. In a multi-trait GWAS of aortic distensibility, *VEGF*, *PDGF* and *IGF* signaling pathways were found to be associated with changes in aortic stiffness [83], all of which were previously identified for their dysregulation in brain vasculature in AD. The same study also found associating variants in well-known Mendelian CVD genes *MYH7* [381], *TBX20* [382], and further recapitulated *LOX* [83,209,380,383]. *VEGF* protein levels, measured in plasma, were also independently associated with both AD patients and those with a history of CVDs in a study by Theeke and colleagues [360]. *MMP9*, a marker of vascular inflammation elevated in AD conditions, is linked to other vascular disease conditions like aortic aneurysm, vascular stenosis, and vascular calcification [209,212,332]. Donepezil, a drug traditionally used to treat AD but tested in cardiovascular phenotypes, reduced levels of *MMP9* and has been observed to reduce the risk of future adverse cardiac events [384,385]. Additionally, *IGF-1* is thought to play a role in Aβ clearance pathways, decreasing the buildup of the protein. *IGF-1* is downregulated in the cerebrovasculature in AD, propagating further toxic aggregation. Increased TGF-β signaling (components LTBP4 and HIPK3) has also been genetically associated with increased aortic stiffness [83]. This is in line with the increase in TGF-β signaling seen in chronic hypertension and aortic diseases [115,281,386,387].

The inflammation and dysregulation of vessel growth and maintenance pathways are hallmarks of both AD progression and arterial stiffening, indicative of increased cardiovascular risk. The breakdown of the BBB instigates a damaging positive feedback loop promoting further breakdown, including the unregulated exchange of molecules and proteins, like Aβ, to move between the parenchyma and the circulatory system. Once able to diffuse into the vasculature, these toxic proteins can adhere to the vascular wall, further hindering the function of the BBB. In the next subsection, we will further describe the ramifications of this protein circulation.

##### Circulation of Metabolites and Protein Deposits

A number of studies have used GWAS- and MR-based methods to identify circulation markers in connection with CVD phenotypes and AD [65,359,361,362,363,365,366,388]. For example, *APCS* encodes serum P amyloid component (SAP) which stabilizes and promotes the aggregation of Aβ [388]. Using variants within the cis–gene region of *APCS*, MR analyses detected a causal association of SAP with coronary artery disease and AD, though only to a marginal degree [388]. Levels of plasma homocysteine seem to have a genetic component [365,389,390], and high levels have also been associated with both stroke and AD independently of other vascular risk factors [364]. The largest connection, however, appears to be in the circulation of Aβ and hTau. The accumulation of Aβ plaques is believed to begin decades before AD onset [352]. Abnormal phosphorylation of tau can similarly be measured up to 20 years before AD onset [391]. *APOE*, the major risk factor gene for late-onset AD, is believed to play a central role in regulating Aβ clearance pathways [349,392], with the slowest clearance observed in *APOE4* mice [392].

The dysregulation of epithelial cells within vasculature and overall breakdown of the BBB described in AD and CVDs results in the diffusion of proteins from the brain into the circulatory system and vice versa. Once in the circulatory system, these proteins are free to deposit in vessels and organs, expanding their toxic impact. The overexpression of neuronal Aβ in mouse models has exhibited damage signatures in cerebrovasculature [324] like microbleeds [351,393]. Greater Aβ and vascular damage have also been associated with greater accumulation of tau [351]. AD cases carrying the *APOE4* isoform have greater amounts of Aβ in cerebral blood vessels [162,394], but have about same amount of cerebral parenchymal Aβ, indicating that though Aβ may not be accumulating at a greater rate, more is able to cross the BBB into the vasculature [162]. Higher blood Aβ levels have been found in both AD patients and those with CVDs [13]. Even without prior CVD diagnosis, plasma concentrations of Aβ and pTau were higher in patients with proteomic indicators of cardiovascular risk or HF [198]. Increase in blood plasma Aβ is also reportedly associated with a greater risk of HF, though this only the case for men [319]. In a GWAS conducted by Sarnowski and colleagues for circulating tau, a significant association was found in *MAPT*, known for its association with AD. Another locus previously known for its association with ischemic stroke was also identified, and other results were enriched for genes reported with AD [355].

In addition to increasing Aβ and tau deposition within circulation, the deposition of protein aggregates exist in the heart in case of both AD and HF [13,184,194,197,198,199,200,201] (Table 4). The same has been found in idiopathic dilated and hypertrophic cardiomyopathy [184,199,348], even without any indicators of cognitive impairment [395]. A cohort of cognitively healthy, aging individuals from a study by Johansen et al. also confirmed an association of elevated Aβ levels in the brain with impaired cardiac atrial function [298]. Cofilin-2, an actin protein involved in aggregation, has specifically been implicated in cytoplasmic aggregates found in both AD brains and the myocardium of idiopathic dilated cardiomyopathy [200,396]. Cofilin-1 has also recently been associated with AD in genetically predicted plasma protein levels [358]. This trend of aggregate deposits further extends to hTau, which aggregate in AD brains, as well as myocardium in HF [197]. HF may also lead to an increase in the process of phosphorylating tau [397]. It is believed that increased levels of Aβ and tau in myocardial tissue negatively impact cardiac function [13,296], and may explain subclinical cardiac changes including those observed in AD [194]. AD patients often similarly present with diastolic dysfunction [13,117,194], decreased LV end-systolic volume, and a lower maximum VO2 [354], even when free from known heart disease [194,354]. Similar worsening of cardiac function has been seen in hTau mouse models [197], where cardiac function continued to decline alongside progression over time.

## 4. Discussion

In summary, the epidemiological connection observed between CVDs and AD is impacted by overlapping genetic mechanisms, as evidenced by multiple studies utilizing statistical genetics methods. While studies of increasing cohort sizes and of greater ancestral diversity are necessary to draw lasting conclusions, the studies presented here offer a foundational starting point for investigation. Genome-wide methods, mainly GWAS and genetic correlation analysis, have provided mostly positive evidence for genetic similarity between AD and CVDs (Table 1), though further studies are necessary. These large-scale methods may be limited by the complexity of AD and CVDs themselves, making cohort generation from non-disease-specific biobanks like the UK Biobank [398] more challenging. This limitation also appears for MR studies, where the selection of instrumental variables is based on existing summary statistics. This was perhaps the most common type of analysis reviewed within our scope. Studies focused on specific sets of genes or variants were able to draw more concrete conclusions, and transcriptomic analyses of tissue derived directly from the vasculature of patients with AD or CVDs provided the most accurate representation of disease conditions. From the results of this literature search, we report the genetic effects believed to contribute to the progression of AD and CVDs. Finally, we discuss and summarize our findings and propose directions for future research in this domain.

At the individual gene level, specific families of genes seem to have the most penetrant and widespread effects (Table 1). First, the familial AD-risk genes *PSEN1* and *PSEN2* result in increased cerebral Aβ levels and promote functional impairment [120,123,124,125,126,182,184]. Second, *APOE*, a major risk factor for late-onset AD [138,149,186], also appears to impact mechanisms that contribute to both complex diseases [62,95,103,165,207,252], with the *APOE4* isoforms having the most harmful effects [44,138,148,187,202]. *APOE* plays a role in both Aβ metabolism and Aβ load [29,164], as well as in cholesterol metabolism [226,250], including LDL levels.

At the pathway level, high Aβ aggregation negatively impacts brain function by inducing inflammation while simultaneously depositing onto cerebral vessels and causing further widespread damage [142,162,164]. Paired with BBB breakdown, the circulation of Aβ and hTau proteins from the brain to other organs within the circulatory system like the heart begins to hinder myocardial functions and may eventually lead to heart disease and eventual heart failure [15,197,297,397]. Detrimental mutations in genes involved in cardiac function such as *TOMM40* and *BIN1* can also contribute to heart damage [82,205,316,350] (Table 3). Lower cardiac output then negatively feeds back to the brain, as lower blood flow adversely effects glymphatic clearance [202,399]. In parallel, altered cholesterol transport and synthesis regulated by *APOE* and other genes like *PCSK9* results in greater circulating levels of LDL and HDL and altered TGs [158,252,258,328] (Table 2). Without intervention, these molecules are prone to lodging onto vessel walls, creating their own plaques, and constraining blood flow, putting patients at greater risk for CVDs like ischemic heart disease and coronary artery disease. In the most severe cases, the full blockage of crucial arteries and vessels to the heart prompts heart attacks or stroke.

Initial contributory changes can occur decades before noticeable symptoms in the etiology of both CVDs and AD, including the appearance of Aβ plaques and tau in the brain [5,352]. They also aggregate in the myocardial tissues of patients with or at risk of heart disease without cognitive decline [395]. This contributes to BBB breakdown, inciting inflammatory feedback loops which increase oxidative stress and further impair BBB function (Table 4). The same homeostasis and vascular growth pathways, notably *VEGF*, *PDGF*, and *IGF*, seem to be dysregulated in independent transcriptomic measures of vascular tissue of AD cases and in cases of high-risk vasculature associated with low aortic distensibility [83,207,344]. Increased blood pressure also negatively contributes to increasing dysregulation associated with disease progression, especially when seen in midlife [106]. This increase in blood pressure may be due in part to environmental or lifestyle factors, such as inactivity or smoking habits [229,400]. It may also be impacted by genetics through the activity of *ACE* and the angiotensin–renin system [85,282] or increased hyperlipidemia influenced by *APOE* status [250] (Table 3). In either case, increased blood pressure interacts with genetic effects to alter gene expression and further worsen CVD and AD outcomes.

The rise in imaging derived phenotypes (IDPs) presents a novel opportunity for understanding the connections between genetics and subclinical changes in organ structure and function along the course of disease progression. IDPs are quantitative measures extracted from multi-model images like magnetic resonance imaging (MRI), with perhaps the largest publicly available source being the UK Biobank [401,402]. IDPs have so far been used to better characterize disease progression and to find genetic associations though GWAS, as demonstrated by several studies included in this review [32,83,111,204]. However, few have considered brain and cardiac IDPs together in disease-specific cohorts. This type of data, paired with additional layers of information from biobanks like genotype, vitals, and so on, could be a powerful way to expand on the current knowledge of shared genetics in AD and CVDs.

### Strengths and Limitations

This scoping review represents a comprehensive overview of the current state of known genetic overlap between CVDs and AD. While other studies have analyzed the epidemiological overlap and co-occurrence of dementias and CVDs previously, few have focused specifically on the underlying genetic factors as we do here. We synthesized included articles into broad categories which describe the overall impact of highly important gene families, including *PSEN* and *APOE*. In addition to describing additional genes associated with both AD and CVDs, we have also provided context in which they contribute via mechanisms, providing concrete directions for researchers to explore in future work. To ensure that this scoping review is as exhaustive as possible, we sourced literature from two of the largest databases available, PubMed and Scopus, and placed no date restrictions on articles included. This ensures that we have not missed valuable articles and allowed us to note trends in topics over time, as seen in Figure 3. Based on the topics grouped, we note that across categories there has been an increase in studies focusing on the shared genetics of AD and CVDs since a first publication in 1990. Based on Figure 3B, *APOE* has been an increasingly popular topic throughout the years, though publications have slowed within the past couple years, potentially indicating that study interest has been shifting to other contributing disease factors. Indeed, we see that across the mechanistic categories of Figure 3C, the BBB and Aβ, tau, and circulating metabolites have been the most prominent subjects in the past five years.

Regarding limitations, we recognize that additional mechanisms known to contribute to AD and CVDs did not make it into the scope of this review. For example, impaired glucose metabolism has been discussed within the scope of both AD and CVDs [403,404,405,406]. Sleep apnea has also been implicated in AD [407,408], especially in the context of impaired glymphatic clearance [399] mentioned here, as well as the potential influence of the gut microbiome [409] and viral infections [410]. These may also impact widespread inflammation observed in both AD and CVD etiologies. Additionally, while we did see the dysregulation of extracellular matrix (ECM) genes in several studies with regard to AD and CVDs [83,94,246,333], we did not touch on elastin-derived protein (EDPs), which result from the breakdown of elastin of the ECM due to inflammation and aging [411,412]. These proteins have been reported to contribute to progression of atherosclerosis, and have been observed to induce the production of Aβ in the brain [411,413,414]. Therefore, this topic is another major area worth exploring. Additionally, lifestyle factors are believed to impact AD risk [415]. The same is true for CVDs, where factors like smoking, inactivity, or dietary habits strongly disease influence risk [400,416,417]. However, in this review, we only focused on genetics, which may not cover the full extent of disease cause.

There are also several limitations of the studies included that are worth mentioning. First, the timeline that has been proposed over the course of CVD and AD progression makes studying their genetic overlap difficult. The average age of onset of CVD is around 60 years [418], while late-onset AD appears at around 74 years of age [419]. This difference in the age of onset combined with the potential decades of undetected progression for both diseases makes it nearly impossible to study the full scale of the diseases without following a cohort longitudinally. Many of these genetic studies have only focused on one timepoint, do not include preclinical disease, or do not follow the participant until their death. Given that CVD and AD are both progressive, complex diseases, creating reliable cohorts fully representative of their disease of focus is difficult. AD cannot be officially confirmed without post-mortem autopsy. CVDs include a wide range of disorders in including heart and arterial diseases, often with long-term progression and sudden consequences [2]. This can make them hard to predict but also hard to categorize. Sex disparities also exist between the two diseases. AD is more often diagnosed in women, potentially in part because women tend to live to older ages [420,421]. In contrast, men have higher risk of CVD at a younger age, though the overall lifetime risk is similar across the sexes [422,423,424]. Social factors may also affect how often patients are diagnosed and if they receive preventative care, which biases electronic health records.

In addition to limitations in creating the cohorts of the studies, we also noticed limitations in the methods of analyses. When creating accurate cohorts, it is also difficult to control for environmental confounders such as lifestyle, diet, or other habits. We know that environmental effects on genes are present when studying CVD and AD, and many of the studies considered did not include these as variables or covariates. In studying the genetics of CVDs, some studies did not account for medications as covariates or exclusion criteria. Additionally, most of the genetic studies included here have almost exclusively been limited to participants of European ancestry. This is a common limitation of genetic studies, as publicly available data are predominantly of European ancestry, but it limits the full interpretation of results since conclusions may be different in different ancestral groups [353,356]. CVDs in particular are known to be more prevalent in Black and Hispanic populations [425], though they are underrepresented in genetic study populations [426]. Future studies would benefit from the inclusion of diverse ancestries, or ancestry-specific cohorts of non-European ancestry.

## 5. Conclusions

In conclusion, genetics appear to be an import factor in explaining epidemiological overlap between CVDs and AD. Variation in the highly penetrant genes *PSEN* and *APOE* have many far-reaching impacts across brain and heart tissues that lead to the functional decline of both organs. Common variants in several overlapping mechanisms between the two diseases play a role in promoting disease progression and increasing damage through various feedback loops. However, significant limitations remain in the creation of cohorts to fully represent CVDs and AD and study their genetic overlap.

## Figures and Tables

**Figure 1 genes-15-01509-f001:**
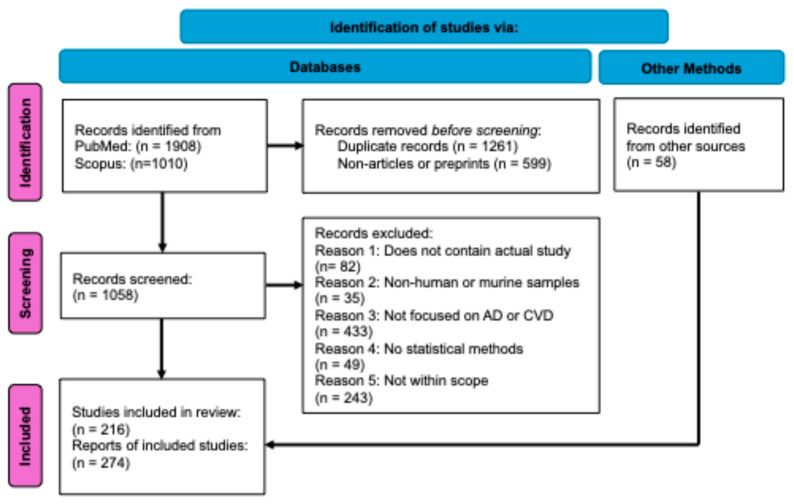
PRISMA flow diagram for scoping review.

**Figure 2 genes-15-01509-f002:**
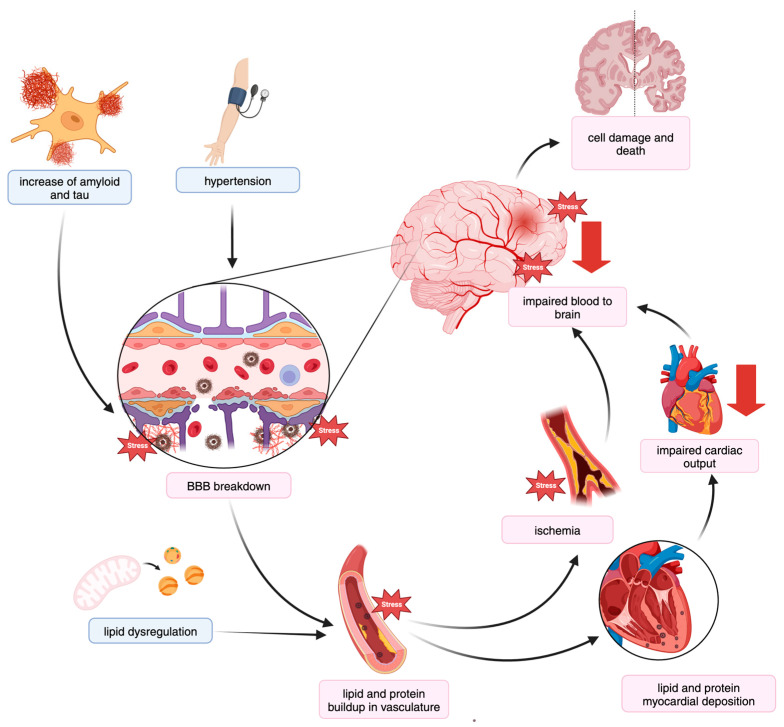
Overview of overlapping mechanisms between cardiovascular and Alzheimer’s diseases (BBB = blood–brain barrier).

**Figure 3 genes-15-01509-f003:**
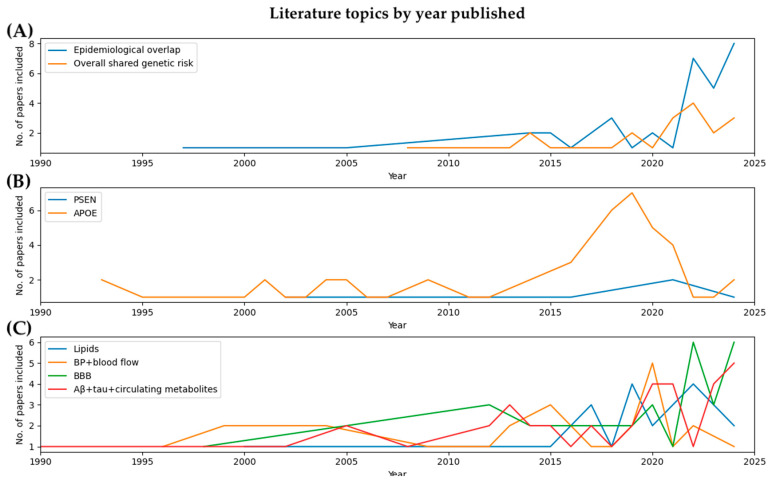
Trends in published literature topics from the year 1990 to the present (2024) comparing CVD and AD etiology. The x-axes describe the year of publication, and the y-axes give the number of papers published in the corresponding year. (**A**) A plot of the published literature topics focused epidemiological and genetic overlap between AD and CVDs. (**B**) A plot of the published literature topics at the gene level describing the effect of familial AD gene families *PSEN* and *APOE* on AD and the brain, as well as CVDs and the heart. (**C**) A plot of the published literature trends relating to the overlapping mechanisms described between AD and CVDs: altered lipid metabolism, changes in blood pressure and their effects, and the breakdown of the BBB and the resulting circulation of proteins.

**Table 1 genes-15-01509-t001:** Shared genetics of cardiovascular diseases and AD (GWAS = genome-wide association study; EWAS = exome-wide association study; ML = machine learning; PRS = polygenic risk score; CAD = coronary artery disease; BP = blood pressure; HR = heart rate; MI = myocardial infarction; CeVD = cerebrovascular disease; MR = Mendelian randomization; MS = mass spectrometry; HF = heart failure; CHD = coronary heart disease; LV dysfunction = left ventricular dysfunction).

Topic	Area of Focus	Articles
General	AD genetics	GWAS [37,38,39,40,41,42,43,44,45,46,47,48,49,50], pheWAS [58,59], methylation [60,61], EWAS [51,52,62], systems [63,64], PRS [65], other [66]
	CVD genetics	Multi-trait [62], CAD [67], BP [68,69,70,71,72,73,74,75,76,77], stroke [53,54,78], MI [55], CeVD [79], cardiac structure [80,81,82,83]
	Overlapping genetics	Association [59,84,85,86], pathways [66], microarray [87,88], MS [88], RNAseq [89,90], systems [89,91,92,93,94], correlation [93,95], MR [96,97,98], gene [99,100,101],
	Effect of CVD on AD and dementia	PRS [31,33,34,102], association [19,20,21,26,31,32,103,104,105,106,107,108], MR [97,109,110,111,112,113,114], multi-omics [115], other [14,17,22,24,27,28,116]
	Effect of AD on CVD	Other [25,116], MR [97,113,114,117,118],
PSEN	AD	[119,120,121]
	CVD	[122,123,124,125,126,127]
APOE	AD	GWAS [49,61], PRS [128,129], EWAS [130,131], multi-omics [132], association [133,134], other [135,136,137,138]
	CVD	CAD [138], stroke [139,140,141], atherosclerosis [142], HF [143], CHD [141,144,145,146], MI [147], LV dysfunction [148], general [133,136,149,150,151,152,153]
	Effect on BBB	[154,155,156]
	Effect on lipids	[59,141,145,146,157,158,159]
	Effect on amyloid	[150,160,161,162,163,164]
	Effect on blood pressure	[165,166]
	Effect on blood flow	[167,168,169]
	AD onset	[170,171,172]
	Inflammation	[173]

## Data Availability

Tables of the studies included in this review can be found in the Appendix A.

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
