# Peer review of "Is the Relationship Between Cardiovascular Disease and Alzheimer’s Disease Genetic? A Scoping Review"

_genes, 2024, doi:10.3390/genes15121509_

Round 1

Reviewer 1 Report

Comments and Suggestions for Authors

The review article titled “Exploring the relationship between cardiovascular disease and Alzheimer’s disease: is it genetic? “ (genes-3288662), is presented in the section "Molecular Genetics and Genomics" of the special volume "Genomics and Genetics of Cardiovascular Disease", and its theme aligns well with both the section and the special issue.

This paper reviews the relationship between cardiovascular disease (CVD) and Alzheimer's disease (AD), two conditions that are highly prevalent in ageing populations and often co-occur. The aim of the article is to explore the relationship between these two diseases, which remains uncertain. Epidemiological studies suggest that CVDs increase the risk of AD and vice versa. Furthermore, both diseases have a high individual heritability (>50%), indicating that genetic variation plays a significant role in the onset and progression of these conditions.

Regarding the title, it is appropriate for the content of the paper but could be more informative by explicitly indicating that it is a review.

The abstract is well-structured, but it does not specify the type of review conducted or the time frame covered. These two elements should be included, as the abstract is the only section that is freely available across all platforms and is key to disseminating the content of the paper.

The introduction highlights the global significance of both diseases, and the references used are appropriate. However, the final part of the introduction presents the hypothesis, but from lines 69 to 74, the objectives are intertwined with the methodology. I suggest rewriting the objectives in a clear and precise manner.

In the materials and methods section, as with the abstract, the type of review is not defined. Although it appears to be a comprehensive review, I recommend that the authors explicitly state the type of review undertaken. The paper follows PRISMA guidelines; however, PRISMA recommends reviewing at least three databases, whereas only one is used here. It is crucial to specify the time period of the literature reviewed, the keywords used in the search, and the inclusion and exclusion criteria in more detail, as the current description merely states “non-relevant” without further explanation.

The results section is well-structured, making the findings easier to understand. The inclusion of a highly illustrative figure is commendable, but, if possible, it would be beneficial to add tables summarizing the main findings of the reviewed studies. As there are many studies, the paper provides very little information on each, and tables would help organize the key results. Overall, the work is impressive and aligns with the initial objectives.

The discussion synthesizes the findings and places them in the context of current knowledge, and I consider this section to be the most engaging part of the paper. The authors reflect on the limitations, but I believe they should also discuss the strengths of the study to provide a balanced evaluation.

The conclusion aligns with the results obtained and addresses the objectives set out.

Reviewer 2 Report

Comments and Suggestions for Authors

Review of the manuscript entitled: Exploring the relationship between cardiovascular disease and Alzheimer’s disease: is it genetic?

The manuscript touches on a very important problem of neurodegenerative diseases, especially AD. Currently, the number of AD cases is constantly increasing, so it is important to determine its causes, which will help prevent new cases. Overall, the manuscript is interesting but some corrections should be made before publication.

1.      The abstract and introduction are prepared mostly correctly. From technical notes, the aim of the manuscript should be clearly indicated in the abstract and at the end of the introduction e.g. "The aim of the present study was to ...".

2.      The methodology is described properly.

3.      The manuscript is devoted to the link between genetic causes of AD associated with incident cardiovascular diseases. Currently, many works from Europe and China show that elastin derived peptides (EDPs) entering the brain during stroke and CVDs are causes of AD. I think it should be mentioned, at least in the introduction. In particular, the authors in FIG 2 describe the disruption of the BBB. Therefore, describing the role of EDPs seems to be important.

4.      All acronyms and abbreviations should be explained when they are first used e.g. line 100 “GWAS” and similar, check the entire manuscript carefully. Others that I found “CAD”, “MR”, “HF”, “LD” and so on.

5.      The description of the results is very good. Good job! To improve readability, maybe add a summary table with references and a very brief collection of the most important information.

6.      A big advantage of the manuscript is the presentation of limitations. Presenting what should be done next is a very good practice.

Reviewer 3 Report

Comments and Suggestions for Authors

Apolipoprotein E4 allele is identified as a risk factor for both Alzheimer's Disease and cardiovascular disease; this review is in place. However, for a scoping review, it is essential to search 2 to 3 databases. Relying on just one database, as currently done, introduces significant methodological shortcomings. Once the necessary corrections are made, I will gladly perform the review again.

When citing references in text, add a space between the text and citation (e.g., xxxxxx [x]).

In line 40, is the percentage 30%, 50%, or 60%?

Please define ( h ) in the context of  ln 46.

Round 2

Reviewer 1 Report

Comments and Suggestions for Authors

Thank you very much for allowing me to review once again the article titled “genes-3288662_Is the relationship between cardiovascular disease and Alzheimer’s disease genetic: a scoping review.”, as well as the authors' responses to the suggestions provided to enhance the understanding of their work.

In the new version uploaded to the platform, the changes made are not clearly indicated, making it challenging to review the document.

In the abstract, it is essential to specify that this is a scoping review covering the period from 1994 to October 2024, which includes the 274 articles reviewed. This is a fundamental detail to ensure the study can be effectively linked to future reviews.

The objective should clearly state what the review aims to clarify. It must be precise and distinct, without conflating the objective with the methodology, as these are different concepts. Therefore, I suggest that the introduction concludes with a clear and precise articulation of the objective. In the methodology section, the process followed to achieve this objective can be detailed.

According to PRISMA (Preferred Reporting Items for Systematic Reviews and Meta-Analyses), there is no strict requirement for the number of databases to be reviewed in a systematic review. However, it recommends searching multiple relevant databases to ensure comprehensive coverage and minimise the risk of bias. Generally, it is advised to include at least three major databases pertinent to the study's subject. For example, in health and medicine topics: PubMed/MEDLINE, Embase, and the Cochrane Library are often recommended.

This recommendation aligns with those from PROSPERO, the international registry for systematic review protocols, which does not mandate a specific number of databases but expects rigorous and well-documented searches to meet high standards like those outlined in PRISMA. A minimum suggestion is to review at least three relevant databases for the field of study, ensuring sufficient coverage and reducing the risk of bias.

Since the authors have reviewed only two databases, this study will be considered a scoping review.

From lines 102 to 108, the procedure is described; therefore, it does not belong to the results section. I suggest relocating this content to the materials and methods section, as it pertains to the methodological process of the study. The sentence found on lines 108 and 109 should instead be moved to the discussion section.

Table 1 should be placed alongside its explanation in Section 3.1.

The new Table 2 provides very interesting information, but the authors could elaborate further on whether the studies indicate positive, negative, or non-significant associations. The same suggestion applies to the other tables.

Reviewer 3 Report

Comments and Suggestions for Authors

This version is much improved, and I have no further misgivings, except that the list of works cited is still outdated. Authors are allowed to use a few references that are a decade or two old, but most references should be published within the last few years. I disagree with the author's point that the two included databases "should be more than sufficient"—anything less than three databases is inappropriate.
